# Behavior of Nymphs and Adults of the Black-Legged Tick *Ixodes scapularis* and the Lone Star Tick *Ambylomma americanum* in Response to Thermal Stimuli

**DOI:** 10.3390/insects13020130

**Published:** 2022-01-26

**Authors:** Fernando Otálora-Luna, Joseph C. Dickens, Jory Brinkerhoff, Andrew Y. Li

**Affiliations:** 1Department of Biology, School of Art & Sciences, University of Richmond, Richmond, VA 23173, USA; joseph.dickens@richmond.edu (J.C.D.); jory.brinkerhoff@gmail.com (J.B.); 2Invasive Insect Biocontrol & Behavior Laboratory, USDA, ARS, Beltsville, MD 20705, USA; Andrew.Li@usda.gov

**Keywords:** thermal biology, ethology, sensory ecology, convection heat, radiation heat

## Abstract

**Simple Summary:**

Ticks use heat emitted by warm-blooded animals to orient to potential hosts for blood-feeding. Here we study the behavior of two species, the black-legged tick and the lone star tick, to sources of heat emissions representative of a human host. First, we offered a heat source to females walking on a locomotion compensator (servosphere). While speed, walking distance, displacement and linearity were unaffected by the source of heat, walking trajectories of ticks were aimed toward this thermal source. In a double-choice walking bioassay, both nymphs and adults of both sexes of the lone star tick, but not the black-legged tick, oriented to a thermal source. From a practical standpoint, a source of heat might be used in combination with other, e.g., chemical or visual, signals in strategies aimed at the survey or control of targeted tick species.

**Abstract:**

Ticks use chemical and thermal signals emitted by humans and other vertebrates to locate suitable hosts for a blood meal. Here, we study the behavior of black-legged *Ixodes scapularis* and the lone star ticks *Amblyomma americanum* exposed to heat sources held at temperatures near those of vertebrate hosts (32 °C). First, we used a locomotion compensator to test behavioral responses of ticks to an infrared light emitting diode (LED). The servosphere allowed us to measure parameters such as velocity, acceleration, linearity, and orientation. Then a heating element (Peltier) located in one of the extremes of a double-choice vertical rod (flying T), was employed to observe upward movement of the ticks toward such a heat source. While both species oriented toward the LED, only lone star ticks were attracted to the Peltier element while climbing upward. In conclusion, we showed that heat attracted ticks from short distances up to several centimeters on a the servosphere, and those responses differed between the two species of ticks on the flying T. We discuss our results in the context of the ecology of both tick species and their potential in tick survey and management.

## 1. Introduction

The black-legged tick *Ixodes scapularis* and the lone star tick *Amblyomma americanum* are hard tick species (Acari: Ixodidae) that parasitize vertebrates including white-tailed deer and humans. The distribution of both species is expanding in North America with the subsequent geographic expansion of tick-transmitted epidemics [1,2,3]. For example, we have observed that certain genetically discernible *I. scapularis* populations—more closely related to northern populations from Massachusetts, New York, etc. than to Virginian coastal tick populations—are dramatically increasing at high elevations in southern Appalachian Mountains, in locations where Lyme disease is rapidly emerging [4,5]. Similar biogeographic redistributions of *I. scapularis* and *A. americanum* are occurring at diverse latitudes and longitudes in North America [6]. The dynamics of this expansion is due mostly to climate and landscape changes produced by anthropic activities [7,8].

Both the black-legged tick and the lone star tick are three-host ticks, so each parasitic stage (larva, nymph and adult) normally feeds on a different individual host in a life cycle spanning 2 to 3 years on average [9]. Adults of both species are commonly found on deer. The immature *A. americanum* are typically found on deer and less often parasitize small and medium vertebrate mammals and birds. Immature *I. scapularis* has a much broader host range [10] as they are commonly found on smaller mammals (e.g., white-footed mice, foxes, raccoons, rodents, etc.), reptiles [11], birds [12] and even amphibians [13]. Agents of Lyme disease and anaplasmosis are spread to humans by the black-legged tick, while agents of ehrlichiosis, Rocky Mountain spotted fever and Heartland virus disease, among others, are transmitted by the lone star tick, which also induces the increasingly prevalent red meat allergy in humans [14].

Ticks searching for a blood meal use chemical, visual, olfactory and thermal signals to locate potential hosts. They are attracted to olfactory cues emanating from hosts including CO_2_, NO, acetone, octenol as well as certain carboxylic acid, phenol and indole volatile molecules [15,16,17,18]. Ticks possess a keen chemosensory ability as most tick species are without eyes except for a few genera that possess primitive paired eyes (i.e., *Dermacentor*, *Hyalomma*, *Amblyomma*) [19]. While *A. americanum* is equipped with primitive eyes [20], *I. scapularis* presumably possesses a rudimentary visual system consisting of lenses fixed over a small field of photoreceptors called ocelli that are capable of detecting light and possibly distinguishing silhouettes [21], analogous to occelli described in the black-legged tick *Ixodes ricinus* [22]. Ticks also possess Haller’s organ located on the dorsal surface of the tarsi of the forelegs [23], which comprises a capsule and a pit containing an array of sensilla that house sensory receptor cells that detect kairomones (e.g., CO_2_), pheromones [24,25,26], humidity and temperature [27]. The ticks’ Haller’s organ possesses a recently discovered thermo-sensory function. Mitchell et al. [28] as well as Carr and Salgado [29] have recently made behavioral observations vis à vis the role of the Haller’s organ in the attraction of ticks to infrared (IR) radiation (or radiant heat) and convection (or convective) heat.

Heat from warm-blooded hosts provides a thermal reliable cue to blood-feeding animals, including blood-sucking insects [30,31], as well as other animals, e.g., fire beetles [32], viper, python and boa snakes [33], and vampire bats (*Desmodus rotundus*) [34,35]. Since sources of infrared radiation (or thermal radiation) produce convective heat (or thermal convection) as well, it is difficult to separate their effects on animal behavior. There are no natural sources that emit infrared radiation without emitting some convective heat. Thus, we should acknowledge that this study does not attempt to disambiguate between these two sources—i.e., infrared and convective heat—but rather operates under the assumption that ticks are commonly exposed to both combined stimuli in natural conditions, and that if an application is designed to attract ticks in the field, it would be unnecessary and perhaps impractical to separate both stimuli.

The purpose of this work was to study the response of *I. scapularis* and *A. americanum* to heat representative of human skin [36] in two behavioral arenas. First, we used a locomotion compensator (servosphere) to test behavioral responses of ticks to an infrared light-emitting diode (LED). The servosphere allowed us to measure parameters such as velocity, acceleration, linearity, and orientation. Then a heating element (Peltier) located above a vertical rod (flying “T”) was employed to observe upward movement of the ticks toward such a heat source. The rationale behind that procedure was to test two predetermined devices that emit heat, which elicit attraction behaviors in ticks, and to compare ticks’ responses on the aforementioned arenas.

## 2. Materials and Methods

### 2.1. Experimental Animals

Adults and nymphs of *I. scapularis* and *A. americanum* were obtained from the tick rearing facility at the Department of Entomology and Plant Pathology, Oklahoma State University, USA. Adults and nymphs of each species were housed separately in an environmental chamber (Model dros33sd4, Power Scientific Inc., Pipersville, PA, USA) in groups of 5–20 in round plastic containers (height 3.8 cm, diameter 2.2 cm) at 99 ± 0.1% RH, 22 ± 0.5 °C, 12:12 L:D. The light cycle regimen was inverted so scotophase that occurred from 7 am to 7 pm facilitating behavioral observations during this time period as tick species are generally more active in darkness [22]. Ticks tested were starved for 2–5 months after molting, prior to behavioral experiments, to normalize arthropods’ physiological conditions and induce appetitive behaviors. Experiments were undertaken during the early scotophase, for about 1 month and all ticks were from the same cohort. Individual ticks were tested only once. We tested 20 females of each species on the servosphere. We tested 30 females, 30 males and 30 nymphs of each species on the flying T.

### 2.2. Behavioral Arenas

We used two different types of arena to assess the behaviors of both tick species. In order to accurately measure kinematic parameters of ticks walking towards a thermal source, we tested female individuals of each species on the servosphere (*n* = 20 females). The thermal source, used on the servosphere, has proved to be effective by Mitchell et al. [23]. The advantages of using a servosphere is that it allows an unimpeded animal to walk in all directions and to measure its kinematic parameters, as explained in the following paragraph. After elucidating the kinematics of ticks towards such a previously tested thermal source on the servosphere, we exposed both species—including females, males and nymphs—to a more natural scenario, a flying T, where they could climb an object as they do when questing in vegetation. The simplicity of this scenario made it possible to test a higher number of ticks and stages.

*Servosphere*. The walking behavior of individual ticks was recorded on a servosphere or locomotion compensator (LC-300, Syntech, Buchenbach, Germany). An untethered tick was allowed to walk unimpeded in all horizontal directions on the apex of a 30 cm diameter white sphere (Figure 1A) in an open-loop set up [37,38,39]. In an open-loop set up, also called a non-feedback set-up, the stimulus intensity is independent of the insect’s kinematics, i.e., the animal cannot perceive if it is approaching the stimulus source. A movement detector based on active pixel sensor (APS)/CMOS technology produced a video signal which was digitally processed by computer algorithms and sent to low inertia servo-motors positioned orthogonally at the equator of the sphere that moved it in the opposite direction of the tick’s movement, thus maintaining the tick’s location at the apex of the sphere. Information on movements of the sphere was supplied to a computer by two pulse-generator encoders positioned orthogonally at the equator of the sphere. This allowed for reconstruction and analysis of ambulatory movements of the ticks that were measured at a frequency of 0.1 s with a spatial resolution of 0.1 mm. An 8-LED lamp (white light), which was not a relevant heat source, integrated with the CMOS camera was positioned 10.5 cm above the tick on the apex of the sphere. In addition, video records using the servosphere’s CMOS camera (LC-300, Syntech, Buchenbach, Germany) and an additional CMOS camera (EOS Rebel T7i, Canon, Tokyo, Japan) were made of stereotyped behaviors during walking throughout each experiment. An isolation dark box (length 36 cm, height 14 cm, wall thick 5 mm) made of polymethyl methacrylate plastic (Precision Plastics Inc., Beltsville, MD, USA) was placed over the servosphere, but allowing the sphere to rotate freely underneath (Figure 1A). In order to track the tick, the roof of the dark box had a window over the apex of the sphere that coincided with the field of view of the video camera. An individual tick was placed on the servosphere with forceps. The experiment started after observing the tick walking for three minutes to determine if it was walking healthily and allowing it to adapt to the servosphere conditions. As all ticks walked normally, individuals were not discarded. The experimental procedure comprised contiguous recording of a 2 min pre-stimulus period, a 2 min stimulus period in which a thermal source was present, and a 2 min post-stimulus period. The thermal source was a LED flashlight (length 15 cm, frontal wide 4.5 cm, model T20 IR, Evolva, China). The front of the LED lamp was 5 cm from the tick. Infrared light emitted by the LED had a narrow bandwidth range of 45 nm; calculated from the two wavelengths on either side of the spectrum given at the intensity value that equals half of the peak value 850 nm (range limits 830, 875 nm). We turned one LED lamp at least 15 min before behavioral observations. The LED lamp became steadily hot after 15 min, reaching up to 41 °C on its front (Figure 1a) as measured by an infrared camera (E5, Flir, thermal sensitivity <0.1 °C, spectral response 7.5 to 13.0 μm, Wilsonville, OR, USA). Following Wien’s displacement law these temperature values indicate that the LED lamp, considered as a black body (i.e., emissivity 0.95, close to human sensitivity), emitted long-wavelength infrared with a peak at 9.22 μm (range limits 7.5, 11 μm), being able to produce convective heat. As the torch temperature was above ambient temperature it also emitted convective heat [40] (p. 570). As the LED lamp symmetry is complex (Figure 1b,c), there is no simple way to calculate the longest horizontal distance that the convection current extended beyond it, but its order of magnitude was in centimeters (around 5 cm).

In our preliminary analyses we observed that certain errors raised in digitizing body movements that were not engaged in the tick’s locomotion, e.g., pivoting and wobbling while walking, grooming and swaying movements while stopped, as well as turning without translation [41]. Thus, for analysis of tracks obtained from walking on the servosphere, the x-y coordinates provided by the servosphere at intervals of 0.1 s were merged in step-sizes of 10 units for the female ticks for more efficient summarizing of the tracks [42]. This merger provided step size intervals that allowed the tick to move at least a fraction of its body length before recording its next position and was large enough to reduce noise produced by the movements mentioned above. Instantaneous displacement and direction were computed from position changes within each interval. The walking tracks of the ticks were reconstructed by plotting the cumulative addition of consecutive positions. Kinematic parameters such as direction (mean vector angle) with respect to the LED lamp (0°), speed (i.e., magnitude of velocity), walking distance (i.e., track length), displacement (i.e., overall change in position), linearity, cosine of mean direction and upward straightness were calculated from instantaneous values using previously developed equations [41,43]. The mean vector angle of each track was calculated as a circular mean (or angular mean) of instantaneous angles. Each vector was represented in polar plots where the length of each vector was the path straightness [43], a measure of statistical dispersion (of instantaneous angles) that range from 0 to 1. Linearity was calculated by dividing displacement (distance in a straight line from origin to end point) and walking distance traveled along a path. Linearity served to measure tortuosity of the path. Cosine of mean direction provided an index of the effectiveness with which the 0° direction was followed; values for this parameter ranged from +1 (attraction), passing through 0 (non-attraction), to −1 (avoidance). Upward straightness served as an estimate of the efficiency with which the 0° direction (attraction) was followed and it was calculated as the sum of displacements projected on the ordinate (y) axis to the walking distance.

Parameters were compared between control, test (heat source) and end-control 2 min periods (*n* = 20); ticks were stimulated with a heating source (880 nm infrared LED lamp) during the test period. Linearity is an estimate of tortuosity of the path and it is calculated as the ratio of the magnitude of displacement vector to the walking distance. Cosine of mean direction provides an index of the effectiveness with which the 0° direction is followed; values for this parameter range from +1 (attraction) to −1 (non-attraction). Upward straightness is an estimate of the efficiency with which the 0° direction is followed and it is calculated as the sum of displacements projected on the ordinate (y) axis to the walking distance.

*Flying T.* An open T-track dual choice arena (i.e., “T-maze”) allowed individual ticks to climb a ‘‘T’’ then choose between two horizontal directions; at the top of the “T” there were extensions directed upwards at 45° angles [44,45,46]. The vertical segment of the “T” was 6 cm in length; both the horizontal part of the “T” and 45° extensions were 2 cm long. All parts of the “T” were constructed of stainless-steel wire (2 mm diameter). Ticks did not show difficulties climbing this material. A white 9-LED light source (BELZ3AAA-BA, Rayovac) was placed 10 cm above the vertical segment of the “T”. The T was erected by piercing the smallest circular face of a cork stopper (*Quercus suber*; size 3, length 19 mm, top diameter 11 mm and bottom diameter 14 mm). Each tick was placed on the top of the cork. Some individuals were reluctant to climb the “T” for long periods (sometimes hours) after being manipulated, thus a right triangle (10 mm, 30 mm, 31.6 mm) placed on the base of the “T” increased the number of ticks that walked on to top of the cork to climb the vertical segment of the “T”. The triangle increased the surface available for ticks to start climbing. The triangle was made of thin paperboard (0.5 mm) covered with nitrocellulose varnish (Base-top coat, L.A. Colors). The triangle was used for all individuals. The cork was also coated with this varnish. Varnish was used to cover natural porosities and protect the material from chemical contamination.

A source of infrared radiation was provided by a thermoelectric plate (Figure 1B) or Peltier element (width 30 mm, thickness 3.6 mm, VT-127-1.0-1.3-71, TETechnology Inc., Traverse City, MI, USA) that was painted black (9198 matte black, Krylon) in order to decrease its reflectance [47]. Human skin has an emissivity of about 0.95 [48]. According to Planck’s Law the radiation emitted by the warm surface of the Peltier element equals the spectrum profile and intensity of radiation emitted by human skin, which emits radiation as it is a black body [49]. Objects whose absorption closely approaches 100% have emissivities almost equal to 1 and are referred to as “black bodies.” Humans emit radiation in the long-wavelength infrared region as if they were “black bodies.” Since human emissivity is so close to 1, accurate broad-band measurements of such electromagnetic waves emitted by the skin may be converted directly into values of temperature. Accordingly, the surface of the human body is an ideal object for thermography (thermal imaging); human body size and shape can be visualized using natural or artificial thermoreceptors [49]. The temperature of the Peltier element was maintained at 32 °C (Figure 1d) using a DC power supply (382270, Quad output, Extech, Nashua, NH, USA) and monitored using a thermal imaging camera (Flir E5, see above for details). The thermal imaging camera also served to verify the temperature of both arenas, checking for unintended infrared reflections. Emissivity of the Peltier was 0.95 as measured by the imaging camera. According to Wien’s displacement law, 32 °C corresponds to a wavelength peak at 9.50 μm in the IR (range limits 8.5 μm, 11 μm). The element also radiated convective heat [40] (p. 570). Following Bergman et al. [40] (pp. 562–572), we calculated 2–5 cm as the longest horizontal distance that the warm current extended beyond the Peltier element.

The Peltier element was clamped 5 mm from the end of the 45° extension of the “T”, so the center of the element coincided with the end of the extension. The control (Figure 1e) consisted of a similar Peltier element at ambient temperature (24 ± 1 °C) positioned at the other end of the “T”. Control and test positions were rotated every 10 ticks. Behaviors were recorded using a CMOS camera (Canon EOS Rebel T7i, Tokyo, Japan). Experiments were conducted in a darkened room at 35–45% RH in which the only source of light was that associated with the bioassay device. In order to test for asymmetries, an experiment was performed in which room temperature Peltier elements were offered on both sides of the flying T to *A. americanum* nymphs. The flying T was successively cleaned with hexane and dried with antistatic poly-shield wipers (Kimberly-Clark, Irving, TX, USA) before each experiment. Ticks were tested during their scotophase.

### 2.3. Visual Recording

To analyze behavioral details, frontal views of the experimental arenas were filmed full-screen at a resolution of 3–30 frames per second using a video camera (EOS Rebel T7, Canon, Tokio, Japan) equipped with a zoom lens (EF-S 15–55 mm f, IS, Canon, Tokyo, Japan) coupled to a macro lens (close-up +10, 58 mm, Commander) to obtain a field view of 40 mm (in the diagonal). The light source consisted of an LED (2850K, 570 lumens, Reveal, General Electric, Boston, MA, USA). Recordings were analyzed by a computer using video software (version 9.1.2.7, Wondershare Filmora9, Shenzhen, China) for replay, playback, tracking and editing. The experimenter was 2 m away from the tick

### 2.4. Statistics

Kinematic and statistical analyses were performed using a spreadsheet (Excel 2016, Windows) and R [50,51] (version 4.0.3, Vienna, Austria).

*Servosphere*. Parameters were compared between control, test and end-control 2 min periods (*n* = 20). Ticks were stimulated with a heating source (880 nm infrared LED lamp) during the test period. Different letters in Table 1 mean significant difference (Wilcoxon paired test) after post hoc Bonferroni correction (*p* < 0.05). Multiple comparisons of non-circular kinematic parameters obtained on the servosphere were performed by using the Wilcoxon paired test after post hoc Bonferroni correction [52]. A Mardian–Watson–Wheeler test [43,53] was applied to determine whether tick orientations during each period differ significantly from each other; multiple comparisons were performed after post hoc Bonferroni correction. In periods where a particular orientation angle was expected, the data were subjected to a modified Rayleigh test [43]. Since the mean vector angles of the tracks were mostly distributed across the direction of the heat after turning of the lamp (end-control), the corresponding mean angles were doubled [43] (p. 51) prior to applying the modified Rayleigh test. The dispersion of the mean track directions was estimated as the mean angular deviation (s), is expressed in degrees and yields a measure that ranges from 0° to 81.3°. The Wallraff test after Bonferroni correction was applied to compare s values [43].

*Flying T.* Behavioral choices by ticks on the “flying T” were assessed for significant differences by the hypothesis on binomial proportions based on the standard normal approximation (binomial test) [54]. In order to test asymmetries, we tested *A. americanum* nymphs on the arena; when both top-sides of the flying T were provided with Peltier elements at room temperature.

## 3. Results

### 3.1. On the Servosphere

Walking directions differed between the three 2 min periods for females of both *I. scapularis* and *A. americanum* (*p* < 0.05) (Table 1). Speed, walking distance, displacement and linearity did not differ between control, test and end-control period for either of the two tick species. However, the cosines of mean direction and upward straightness were affected by thermal stimulation of both ticks, an effect that remained latent during end-control (Table 1).

Females of both species walked randomly (*p* > 0.05) on the sphere during the 2 min control period prior to exposure to the thermal stimulus (see Figure 2A,B). During exposure to the stimulus, the ticks oriented locomotion toward the IR lamp (*p* < 0.001). During the end-control period, ticks continued to walk toward the thermal stimulus (*p* < 0.005). While angular deviations differed significantly when comparing control vs. test periods (*p* < 0.05), control vs. end-control and test vs. end-control did not differ. Females of both species walked obliquely with respect to 0° (i.e., 45° and −45°) during the end-control (*p* < 0.001). Representative tracks taken by *I. scapularis* and *A. americanum* are shown in Figure 3.

### 3.2. On the “Flying “T”

We tested A. americanum nymphs when both top sides of the flying T were provided with Peltier elements at room temperature to test arena asymmetries and no significant differences were observed between choices made by nymphs (*p* = 0.181, z = 0.91, binomial test). Both A. americanum adults and nymphs (*p* < 0.001) preferred to walk to the 32 °C heating element (*p* < 0.050 females, z = 2.01; *p* < 0.010 males, z = 1.64; *p* < 0.001, z = 3.83 nymphs). However, I. scapularis females (*p* = 0.292, z = 0.55), males (*p* = 0.572, z = 0.00) and nymphs (*p* = 0.292, z = 0.55) did not prefer the heating element (Figure 4). All ticks frequently performed “reaching” by moving their forelegs up and down both during walking and during stops.

## 4. Discussion

Thermal convection and thermal radiation are important cues used by hematophagous animals as they approach a host. The skin of a vertebrate will produce a thermal convection plume due to the air density change caused by its temperature, i.e., the adjacent hot air will be driven to move up by the buoyancy force of the reverse direction of gravity [55], but also will produce convective heat in the form of electromagnetic IR waves (see asterisk in Figure 1A,a). A thermal plume can be affected by wind, while IR is constant and effective from any angle regardless of wind direction and turbulence. Nevertheless, questing ticks are exposed to both thermal stimuli, convective heat (i.e., hot air currents) and radiant heat (i.e., IR electromagnetic waves) emitted by an approaching animal host. From an ethological approach, it is relevant to expose ticks to conditions similar to those encountered in *nature*—in the sense of Whitehead, García Bacca, Otálora-Luna et al. [56,57,58]—as a questing tick will rapidly decide to cling on a host at a short distance from skin. In our experiments, we studied tick kinematics while offering a source that produced both thermal convection and thermal radiation. Our results showed that females of both species studied here depicted trajectories oriented to the thermal stimuli, i.e., near IR (850 nm), long-wavelength IR (~9 μm) and convective heat; while, kinematic parameters such as speed, walking distance, displacement and linearity were not significantly affected by the thermal source on the servosphere. Both, *I. scapularis* and *A. americanum* were attracted to the LED lamp while walking on the servosphere. Speed, walking distance, displacement and linearity did not differ between control, test and end-control period for either of the two tick species, i.e., when exposed to a heat source both species oriented their walks towards the stimulus but without changing their other kinematic parameters. After elucidating the kinematics of ticks on the servosphere we exposed both species, including all their stages, to a semi-natural scenario, where they could climb an object as they do when questing in vegetation. While *I. scapularis* did not show attraction, *A. americanum* ticks were attracted to the Peltier element when walking on the flying T.

The strong attraction behavior depicted by ticks stimulated with the warm LED lamp is not surprising as these cues provide valuable information that facilitates orientation to warm blood hosts. Homeothermic animals maintain a temperature that contrasts with their environment and radiate most strongly in the infrared at a wavelength, between 7.5 and 14 µm (Schmitz et al., 2016). In addition to the importance that thermal information may have in the vicinity of the host, it also might serve to find poikilothermic hosts, mates, oviposition sites, refuge and influence overwintering habitats. Independent from its specific characteristics, every matter with a temperature above the absolute zero point emits electromagnetic radiation, started by molecular movements beginning at 0° K. Certain objects given their microclimate have thermal signs that are large enough to be easily detected through infrared sensing [59] (p. 119). Poikilothermic animals in movement can be detected by integrating infrared stimuli from microhabitats [33].

Earlier reports of ticks responding to heat by Webb [60] and Oorebeek et al. [61], among others, fail to mention that the heat source used in their studies also produced long-wavelength infrared radiation. Mitchell et al. [28] showed that *Dermacentor variabilis* responded to a source of near IR (850 nm) produced by an LED source similar to that used in the present study (see Materials and Methods). These authors demonstrated that the Haller’s organ was responsible for detecting such near IR from a distance of 25 cm. Carr and Salgado [29] suggested that it is likely that these ticks’ IR sensors were warmed by the 850 nm light, which induced thermotaxis, although warming of IR receptors is part of the transduction process [62]. Mitchell et al. [28] failed to mention that the near infrared source used in their study also produced long-wavelength IR, and convective heat, as we showed here (Figure 1A,a). Probably, Haller’s organ was also responsible of detecting the ~9 μm produced by the torch used by Mitchell et al. [28]. Therefore, probably both short and long-wavelength radiations were responsible for the thermotaxis observed by Mitchell et al. [28]. The role of Haller’s organ in far infrared detection is supported by Carr and Salgado [29], who disrupted tick attraction to a 40 °C radiant body by waxing over Haller’s organ capsule apertures. Further studies might focus on the versatility of responses of Haller’s organ to various wavelengths of IR, considering the difficulty of separating infrared from heat stimulus, as long-wavelengths are produced by most thermal sources.

Because both tick species showed strong thermotaxis on the servosphere, we focused on testing a simpler thermal source on semi-natural conditions for subsequent experiments. During host seeking, hard ticks exhibit questing as a way of increasing the chances of coming into contact with a suitable vertebrate, by climbing up a blade of grass or similar plant parts [63]. The Peltier element used here emitted both long-wavelength IR (~9 μm) and convective heat at 32 °C. This plate artificially modeled a warm-blooded host–human skin that emits radiation in the waveband between 4.5 and 11 μm with a peak near 9 μm at 32 °C [64,65]. Carr and Salgado [29] used a similar 10 cm^2^ plate that did not elicit thermotaxis at 22° and 30 °C, but did attract ticks at 37° and 40 °C. Our 32 °C Peltier element attracted both nymphs and adults of both sexes of *A. americanum*, but not *I. scapularis*. We observed that *A. americanum* approached this stimulus from 5 cm; the thermal plume emitted by a human extends horizontally to 2–5 cm from its source, although it can extend a few meters vertically above the head [55,64,65]. Thus, we expect that the thermal plume produced by the Peltier element reached the tick at the junction of vertical and horizontal aspects of the T-track. The scenario represented here not only models human skin, but also the natural questing performed by ticks in the field. A tick will normally climb vegetation to quest for a host and attach to it when the occasion arises, i.e., when the host passes sufficiently close by. In such a situation, a tick will normally be subjected to both IR and convective heat in a range of a few cm. Obviously, other stimuli such as odors, tactile and visual cues could also be involved in the final decision. Moreover, if the tick has chosen an inconvenient target, it may release from the host and begin questing again.

The fact that *I. scapularis* was not attracted to the heat-emitting source while walking on the flying T does not mean that it cannot physiologically detect heat or will not behaviorally respond to it under other circumstances. Behavioral differences between *A. americanum* and *I. scapularis* may be related to their specific innate behaviors. We observed that *A. americanum* is reluctant to walk up, i.e., they tend to exhibit positive geotropism [66]. Perhaps when *A. americanum* does climb up, they are then “prepared” to respond to a thermal source. While *I. scapularis*, which tends to be negatively geotropic [66], may accomplish a different strategic searching step prior to sitting and waiting for a host-emitting heat source.

It is known that ixodid ticks are more active during the less-desiccating hours and in less-illuminated places as saturation deficits limit questing and darkness induces mobility [22]. Ticks climb to the tips of grass and shrubs where they stealthily sit and wait, questing with their forelegs outstretched. Compared to predator arthropods (e.g., spiders, scorpions, assessing bugs, etc.), ticks do not *ambush* (in the strict sense), as ambushing implies violent movements to pursue and kill prey [67]. To avoid dehydrating conditions during questing [68], ticks descend at intervals to the moist litter on the ground where they reabsorb water, a behavior referred by Perret et al. [22] as quiescence. After attaching to a vertebrate host and feeding on its blood for several days, ticks abandon their host and search for refuge and quiescence. While quiescence involves a period of resting, questing involves searching for a suitable place to sit and wait for a passing host. Thus, questing also involves sensing the air with their forelegs, and other sensory organs, in search of chemical, visual and thermal cues emitted by potential hosts. Both species tested here did not feed after molting, thus there is a high probability that they were searching for blood. Both species were attracted to the thermal source on the servosphere, and thus it is surprising that *I. scapularis* did not show interest in the thermal cue on the flying T. This was possibly because *I. scapularis* “needs” to spend more time walking before responding to a thermal source as if it were fulfilling a fixed action pattern [69,70]. In any case, it is not completely unexpected to find such differences between species as each one occupies diverse niches [71], that were not considered in the experimental set up. Their different behaviors can be attributed to the evolutionary fine-tuning of their physiological systems and behavioral adaptations to their different niches over millions of years since the first primitive members of the group appeared [72]. Further studies will be needed to better understand questing behavior in different species of ticks, and how both the sensitivity and specificity of responses to thermal stimuli are involved in it. Such future studies may require a better understanding of tick ecology and tick natural behaviors in order to define more realistic [57] laboratory set-ups and may need to perform field observations, for example, by passing heat sources and controls close to the sitting and waiting ticks.

In previous studies, as well as in our study, only direction of radiation was considered, i.e., direction as a source of information, but electromagnetic waves can be a source of other information that might serve to recognize forms and depths. Heat sources in nature relative to thermal conditions are variable. There is a lack of certainty around what the range of environmental intensities and forms of heat sources that are physiologically recognizable to ticks and how much that influences potential attraction, which limits behavioral research methods. Are ticks able to “see” infrared forms as some snakes do? Are ticks able to perform thermal imaging with their Haller’s organs? A full understanding of the thermal sensory eco-physiology of ticks undeniably requires the study of other behavioral traits vis à vis a larger range of thermal stimuli.

## 5. Conclusions

Our data show that the lamp and Peltier used here, representative of vertebrate skin, attract ticks and that behavioral differences between *I. scapularis* and *A. americanum* must be considered, as each species responds differently to similar thermal stimuli. We show here that a heat source is attractive to ticks from short distances up to several centimeters and this response can differ between two species of tick. From a practical standpoint, a heat source might be used in combination with host attractants and/or environmental odors as a multisensory lure for strategies aimed at the survey or control of targeted tick species. However, further research is needed to better understand the ecological nature of ticks’ responses to IR and convection heat before its usefulness in the management of tick species is ascertained.

## Figures and Tables

**Figure 1 insects-13-00130-f001:**
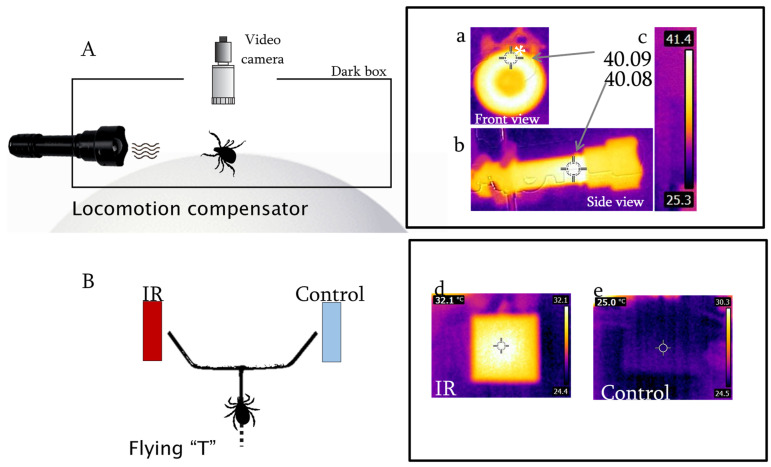
Schematic diagrams representations of the behavioral arenas ((**A**). servosphere, (**B**). flying T) and thermal images of both heating sources (**a**). front view of the infrared (IR) lamp, (**b**). side view of the IR lamp, (**c**). colored scale of the detected temperature range, (**d**). front view of the Peltier element switched on, (**e**). front view of the Peltier element switched off.). Numbers correspond to temperatures in Celsius as measured with the thermo-camera (see Materials and methods for details). The asterisk indicates convective heat flowing up and emitting infrared heat.

**Figure 2 insects-13-00130-f002:**
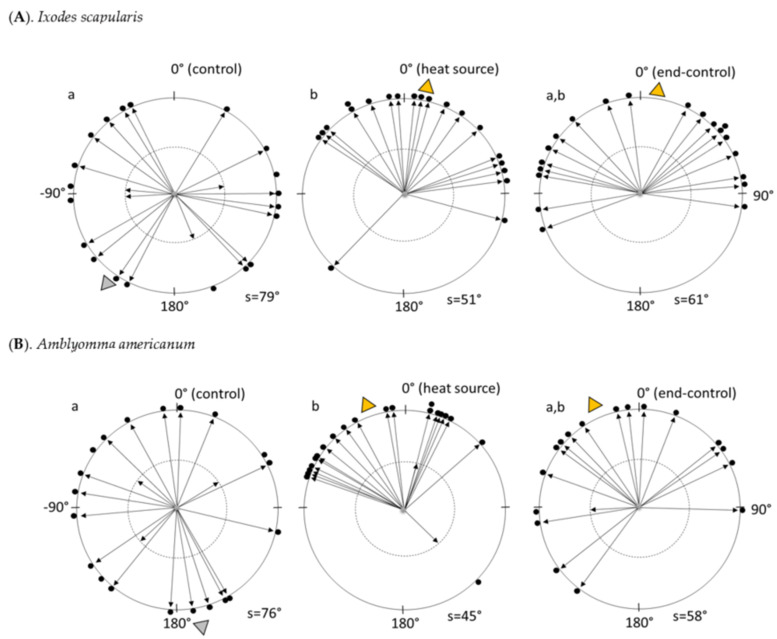
Polar plots of the distribution of the mean vectors (arrows) of tracks described by 20 *Ixodes scapularis* females (**A**) and *Amblyomma americanum* females (**B**) on the servosphere during consecutive 2 min periods (black dots). Parameters were compared between control, test (heat source) and end-control 2 min periods (*n* = 20). Ticks were stimulated with a heating source (850 nm infrared LED lamp) during the test period. Yellow and gray triangles indicate the circular mean of (20) mean vector angles; yellow indicates significantly concentrated angles (Rayleigh test, see Methods). The length of each vector, which is a non-unit parameter ranging from 0 to 1, is a measure of the dispersion of the instantaneous directions taken by a tick during its track. Vectors weighted to the external perimeter indicate that their length was >0.5 and represent straighter walks in that direction. Vectors weighted to the perimeter of the internal circle (dashed line) indicate that their length was ≤0.5. The mean angular deviation (s) below each plot is a measure of the dispersion in degrees of the directions taken by ticks in each period (see Methods). Different letters indicate difference in angular deviation (see Results).

**Figure 3 insects-13-00130-f003:**
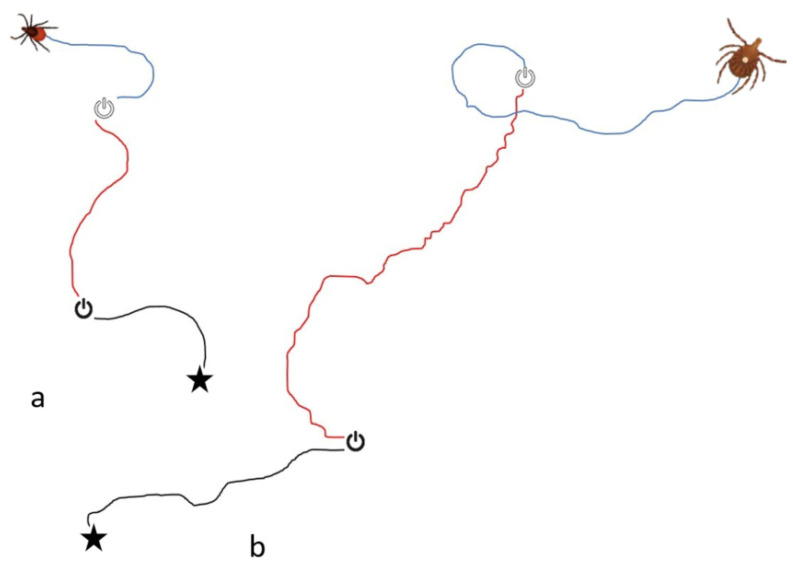
Single tracks made by *Ixodes scapularis* (**a**) and *Amblyomma americanum* (**b**) females on the servosphere. Symbols with a line partially within a broken circle mark onset (black) and cessation (white) of infrared stimulus. The black segment of the track plots the path of the tick during the initial control period with the infrared lamp off. The red segment plots the track during the stimulus period with the infrared lamp on. The blue segment plots the track on the tick following end of infrared stimulus.

**Figure 4 insects-13-00130-f004:**
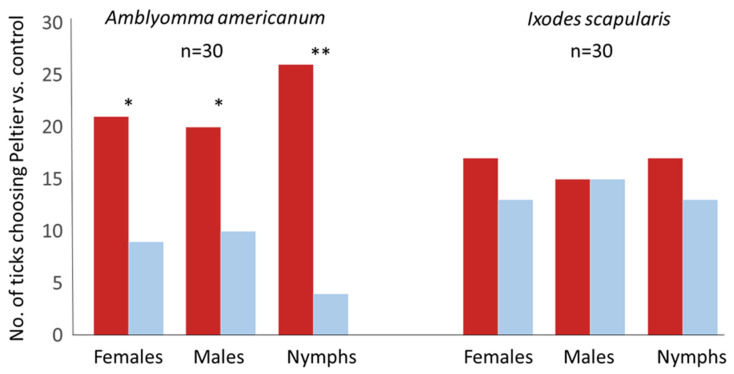
Behavioral responses of female, male and nymph ticks *Amblyomma americanum* and *Ixodes scapularis* to a source of infrared radiation on a flying T-track dual choice arena. The thermal stimulus consisted of a blackbody-like heating element, i.e., Peltier element. Left (red) bars of each pair represent ticks walking to the infrared source, and left (blue) bars represent ticks walking to the control. Behavioral frequencies were compared using the binomial test (* = *p* < 0.05, ** = *p* < 0.001, *n* = 30).

**Table 1 insects-13-00130-t001:** Kinematic parameters recorded on a servosphere where *Ixodes scapularis* (A.) and an *Amblyomma americanum* (B.) females were allowed to walk unimpeded in all directions at the apex of a sphere during 6 min. Parameters were compared between control, test and end-control 2 min periods (*n* = 20). Different letters mean significant difference (Wilcoxon paired test) after post hoc Bonferroni correction (*p* < 0.05), see Materials and Methods. Values without letters indicate *p* > 0.05. LED stands for light emitting diode.

A. *Ixodes scapularis*			
Parameter	Control	Test (LED Lamp)	End-Control
Speed (mm/s)	2.79	3.41	3.31
Walking distance (mm)	296.39	437.77	438.57
Displacement (mm)	129.40	184.24	181.71
Linearity	0.48	0.51	0.54
Cosine of mean direction	−0.16 a	0.51 b	0.61 b
Upward straightness	0.05 a	0.57 b	0.50 b
**B** **. *Amblyomma americanum***			
Parameter	Control	Test (LED Lamp)	End-Control
Speed (mm/s)	5.67	6.80	4.86
Walking distance (mm)	680.61	807.01	579.04
Displacement (mm)	501.25	562.44	451.79
Linearity	0.67	0.67	0.57
Cosine of mean direction	−0.41 a	0.57 b	0.37 b
Upward straightness	−0.27 a	0.45 b	0.15 b

## Data Availability

The data analyzed in this study are available on request from the corresponding author. No new data were created or analyzed in this study.

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
