# Peer review of "Behavior of Nymphs and Adults of the Black-Legged Tick Ixodes scapularis and the Lone Star Tick Ambylomma americanum in Response to Thermal Stimuli"

_insects, 2022, doi:10.3390/insects13020130_

Round 1
Reviewer 1 Report
Manuscript insects 1426610 presents the response of two tick species to sources of radiant heat. Adults and nymphs of the lone star ticks and adults of the black legged tick are guided by radiant heat. The research results provide insight into the ecology of these ticks. The writing is clear and succinct, and the experiments are appropriately described. This is truly elegant research and I think that the readers of the journal Insect would be interested in this study.
Author Response
Thank you for your review of our paper, and for the encouraging comments.
Reviewer 2 Report
The paper by Otálor-Luna and colleagues provides new interesting information on the ecology of ticks important from an epidemic point of view.
my comments and suggestions for consideration by the authors:
First, the paper is not prepared in accordance with the publisher's requirements. In my opinion, it is highly inappropriate to submit an article to a journal without using the valid formatting!
The paper also does not have a Simple summary section
At what physiological age were the tick specimens tested? If in a different physiological age, their response to stimulation could be different.
the description of the statistics used is insufficient. In my opinion, the statistics should be fully described in the new section.
When giving the result of probability, the result of the statistical test should also be given.
Figure 2 is partially invisible, which makes it impossible to evaluate the results.
The paper does not contain a Conclusions section, which is required in the Insects.
Author Response
1. Reviewer 2: The paper by Otálora-Luna and colleagues provides new interesting information on the ecology of ticks important from an epidemic point of view.
R: Thank you!
2. Reviewer 2: my comments and suggestions for consideration by the authors:
First, the paper is not prepared in accordance with the publisher's requirements. In my opinion, it is highly inappropriate to submit an article to a journal without using the valid formatting!
The paper also does not have a Simple summary section.
R: We adapted the format.
3. Reviewer 2: At what physiological age were the tick specimens tested? If in a different physiological age, their response to stimulation could be different.
R: Ticks were tested circa 3.5(+-1.5) months after molting to adults. Individuals did not suck blood after molting and maintained in the same climatic conditions to guarantee the same physiological conditions.
4. Reviewer 2: the description of the statistics used is insufficient. In my opinion, the statistics should be fully described in the new section.
R: We included the new section “2.4. Statistics” to show statistics for each arena, see the subsections Servosphere and Flying T.
5. When giving the result of probability, the result of the statistical test should also be given.
R: We agreed, we showed these values.
6. Reviewer 2: Figure 2 is partially invisible, which makes it impossible to evaluate the results.
R: Figure 2 quality was improved.
7. Reviewer 2: The paper does not contain a Conclusions section, which is required in the Insects.
R: We included a conclusion paragraph after the general discussion.
Reviewer 3 Report
This manuscript describes the attraction/affinity of ticks to thermal/heat sources. It has relevance when considering the monitoring/control of tick populations, particularly in outdoor areas that may be consistently frequented by humans.
The following changes are recommended in order to strengthen the data presented in the manuscript:
Introduction:
-- The authors state that the distribution of tick is expanding in North America. This idea should be elaborated. Describe the dynamics of the expansion.
-- Reference needed after the sentence "They are three-host ticks......years on average).
-- Reference needed after the last sentence of the first paragraph.
-- The second sentence of the second paragraph mentions that ticks are attracted to "other odors". Specify what the "other odors" actually are.
-- If there have been any previous studies detailing the attraction of ticks to thermal sources, those deserve to be mentioned in the Introduction.
Section 2.1:
-- How many ticks were tested (i.e. n = ?). The last sentence states that individual ticks were tested once, however, additional details are needed to clarify the experimental conditions. How many ticks were tested in each experiment? How many were males/females/nymphs?
Section 2.2:
-- Citations to Figures 1 A, B, C, D are not in the text. The text immediately cites to Figure 1E. 1 A, B, C, D should be cited before 1E.
-- Figure 1 needs re-labeled. It is confusing to use 1A, 1B, 1a, 1b, 1c, 1d, 1e.
-- Change "Fig." to "Figure" throughout the manuscript.
-- The first sentence of the second paragraph is not a complete sentence. It should be revised.
-- "Parameters were compared between control, test and end-control...." -- These conditions should be explicitly defined within the text for each experiment.
-- The details concerning the ticks used in all experiments are lacking in the Materials and Methods. As written, the experiments could not be replicated. The following details should be added: number of ticks used for each experiment, number of female/male/nymphal ticks used for each experiment, time of day experiments were individually performed, timeframe of all experiments - were they done in a month, over a year, etc., were all ticks from the same generation. These details should be included for both Ixodes and Amblyomma ticks.
-- Table 1 - Why were females only used for testing?
-- Table 1 - Asterisks should used instead of "different letters" to denote significance.
Section 3.1
-- Specify the control, test and end-control periods for each tick species (first paragraph).
-- Figure 2 - The lettering of this figure should be revised. 2A, 2B, 2a, 2b, 2ab is confusing.
-- Figure 3 - How many ticks/replicated does this figure represent? Is it derived from a single tick?
Section 3.2:
-- Is there data missing from this section? The description seems quite short.
-- Figure 4 - Why were males and females used for this experiment, but earlier only female data was presented? The reasons should be clarified throughout the manuscript.
Discussion:
-- Reference needed after the sentence "A thermal plume can be affected by wind....."
-- Second paragraph - why were only females used?
-- Third paragraph - remove "see Materials and Methods".
-- Third paragraph - was it really shown in this study that long-wavelength IR was produced? If so, is the data included in the results?
-- Third paragraph - Combine the sentences "Probably, Haller's organ...." and "So probably both,....."
-- Fourth paragraph - Human skin was not actually modeled in this experiment. Human skin temperature emissions may have been modeled. But, human skin is more complex than just temperature/heat emission, so stating that human skin was modeled is a large assumption.
-- Fifth paragraph - Change "submitted or in journal review" to "unpublished".
-- Seventh paragraph - Eliminate the questions. These should be made as statements.
Other:
-- Were any ticks from the wild tested? The results may differ between laboratory-reared ticks and wild ticks. Mentioning this in the Discussion would strengthen the manuscript.
References:
-- The font/sizing of the references should match the rest of the manuscript.
Author Response
- Reviewer 3: This manuscript describes the attraction/affinity of ticks to thermal/heat sources. It has relevance when considering the monitoring/control of tick populations, particularly in outdoor areas that may be consistently frequented by humans.
R: Thank you!
- Introduction:
a-- The authors state that the distribution of tick is expanding in North America. This idea should be elaborated. Describe the dynamics of the expansion.
R: Whilst J. Brinkerhoff has worked on this subject, describing the dynamics of the expansion for both species in North America is possibly beyond the scope of our manuscript. However, we agree the idea should be elaborated, thus we briefly showed an example of own work in Virginia to illustrate the current trend in the dynamics of I. scapularis expansion. We included more references as well, following the suggestion of Reviewer 4.
b-- Reference needed after the sentence "They are three-host ticks......years on average).
R: We included the required reference in the first paragraph of the Introduction.
c-- Reference needed after the last sentence of the first paragraph.
R: We included the required reference after the last sentence of the first paragraph of the Introduction.
d-- The second sentence of the second paragraph mentions that ticks are attracted to "other odors". Specify what the "other odors" actually are.
R: We improved this sentence and mentioned the main chemical groups where these odors belong.
e-- If there have been any previous studies detailing the attraction of ticks to thermal sources, those deserve to be mentioned in the Introduction.
R: We had already cited Mitchell et al. (2017) as well as Carr and Salgado (2019) in the Introduction, however, we have improved the redaction, so it is more explicit now.
- [M&M]
Section 2.1: [Ticks]
a-- How many ticks were tested (i.e. n = ?). The last sentence states that individual ticks were tested once, however, additional details are needed to clarify the experimental conditions. How many ticks were tested in each experiment? How many were males/females/nymphs?
R: The number of ticks used on the servosphere was mentioned in legend of Fig 2. The number of ticks used in the T were mentioned in the illustration of Fig 3. Now, these numbers are also included in M&M. We also clarify the rationale behind these numbers.
Section 2.2: [Behavioral arenas]
b-- Citations to Figures 1 A, B, C, D are not in the text. The text immediately cites to Figure 1E. 1 A, B, C, D should be cited before 1E.
R: The corrections was made.
c-- Figure 1 needs re-labeled. It is confusing to use 1A, 1B, 1a, 1b, 1c, 1d, 1e.
R: The comprehension of these figures improved now that they are cited correctly throughout the text. Thank you.
d-- Change "Fig." to "Figure" throughout the manuscript.
R: We changed it.
e-- The first sentence of the second paragraph is not a complete sentence. It should be revised.
R: We improved that sentence.
f-- "Parameters were compared between control, test and end-control...." -- These conditions should be explicitly defined within the text for each experiment.
R: We made this explicit in Table 1, Figure 2 and in other places through the text.
g-- The details concerning the ticks used in all experiments are lacking in the Materials and Methods. As written, the experiments could not be replicated. The following details should be added: number of ticks used for each experiment, number of female/male/nymphal ticks used for each experiment, time of day experiments were individually performed, timeframe of all experiments - were they done in a month, over a year, etc., were all ticks from the same generation. These details should be included for both Ixodes and Amblyomma ticks.
R: We clarified this in the text.
i-- Table 1 - Why were females only used for testing?
R: We clarified this in the text.
j-- Table 1 - Asterisks should used instead of "different letters" to denote significance.
R: In order show the statistical values resulting from making multiple comparisons is very common to use letters instead of asterisk.
- [Results]
Section 3.1: [On the servosphere]
a-- Specify the control, test and end-control periods for each tick species (first paragraph).
R: We made this explicit in the first paragraph of the Results as well as in Table 1, Figure 2, etc.
b-- Figure 2 - The lettering of this figure should be revised. 2A, 2B, 2a, 2b, 2ab is confusing.
R: The comprehension of Fig 2 was improved. Please note that letters a and b here are showed to indicate statistical differences. This method of indication is widely used in the literature when multiple comparisons are statistically illustrated. Different letters (i.e. a and b) indicate difference in angular deviation (see Results).
c-- Figure 3 - How many ticks/replicated does this figure represent? Is it derived from a single tick?
R: We tested 30 females, 30 males and 30 nymphs of each species on the flying T. This is clarified in the legend.
Section 3.2: [On the “flying T”:]
d-- Is there data missing from this section? The description seems quite short.
R: There is not missing data; compared to the servosphere this arena produces less data, but we can run a higher number of tests, and have a realistic view of the attraction effect in a shorter period.
e-- Figure 4 - Why were males and females used for this experiment, but earlier only female data was presented? The reasons should be clarified throughout the manuscript.
R: The rationale behind this is clarified at the beginning of section 2.2.
- [Discussion:]
a-- Reference needed after the sentence "A thermal plume can be affected by wind....."
R: The reference, associated with such statement where we say that a thermal plume –of convective heat– can be affected by wind, while IR is obviously constant and effective from any angle regardless of wind direction and turbulence, was included: Bergman et al. (2014). Please note that infrared energy travels at the speed of light (299 792 458 m / s) without heating the air it passes through, (the amount of infrared radiation absorbed by carbon dioxide, water vapor and other particles in the air typically is negligible) and gets absorbed or reflected by larger objects it strikes.
b-- Second paragraph - why were only females used?
R: This was answered above.
c-- Third paragraph - remove "see Materials and Methods".
R: We deleted it.
d-- Third paragraph - was it really shown in this study that long-wavelength IR was
produced? If so, is the data included in the results?
R: Figure 1a-e, represent such data; these four images were constructed with infrared sensors. Please note that the LED lamp was measured by an infrared camera thus following Wien's displacement law the measured temperature values indicate that the LED lamp, considered as a blackbody (i.e. emissivity 0.95), emitted long-wavelength infrared with a peak at 9.22 μm (range limits 7.5, 11 μm) following Bergman et al. (2011). Additionally, it also induced convection. This is mentioned in Materials and Methods and clarified in the Discussion.
e-- Third paragraph - Combine the sentences "Probably, Haller's organ...." and "So probably both,....."
R: We combined both sentences.
f-- Fourth paragraph - Human skin was not actually modeled in this experiment. Human skin temperature emissions may have been modeled. But, human skin is more complex than just temperature/heat emission, so stating that human skin was modeled is a large assumption.
R: We omitted the word “modeled”.
g-- Fifth paragraph - Change "submitted or in journal review" to "unpublished".
R: The paper has just been accepted for publication in Journal of Ethology, and we might have the reference ready soon.
h-- Seventh paragraph - Eliminate the questions. These should be made as statements.
R: We like the “questions” style of such statements.
- [Other:]
a-- Were any ticks from the wild tested? The results may differ between laboratory-reared ticks and wild ticks. Mentioning this in the Discussion would strengthen the manuscript.
R: All the ticks were lab-reared; this was mentioned in M&M.
References:
b-- The font/sizing of the references should match the rest of the manuscript.
R: Thank you.
Reviewer 4 Report
The manuscript written by Otalora luna et al. is interesting and nicely written.
I am unable to provide a sufficient review for some of the methods and calculations relating to tick orientation. Additionally, no line numbers were included in the draft provided, therefore, I have not matched line numbers with suggested edits. I strongly recommend that authors revise commentary on I. scapularis behavior in the Discussion.
Intro - Need to cite a reference for expansion of I. scapularis in North America… Eisen or Ogden, or numerous others. This is occurring both northward and westward.
Re: Amblyomma americanum and tularemia….yes, a vector in a few extremely rare instances. Practically, I would suggest replacing this pathogen/disease with Ehrlichia ewingii, Rickettsia or Heartland Virus.
- americanum is missing an ‘I’ (spelling in the paragraph preceding Methods section
Methods 2.1 – see formatting gap
Figure 2A is partially obscured, making it impossible to review
Figure 3 – does this represent the track of a single tick? If not, the number should be listed in the Legend. Is this figure mentioned in the text? If not, it’s purpose for inclusion should be stated.
One key element the authors may want to address is the lack of certainty around what the range of intensities of heat source that are recognizable to the tick and how much that influences potential attraction. However, heat sources in nature relative to thermal conditions are also variable, so this is just a consideration and limitation of these methods, but not a fault.
The results that linearity was not affected is surprising and suggests that ticks, at least the more ambulatory Amblyomma were not positioned for enough from the stimuli?
Dermacentor variabilis misspelling in discussion
These two tick species aren’t really known to “drop” on hosts as stated in the Discussion. In nature, they clasp on to hosts passing in close proximity. They do not ascend vegetation higher than the height of vertebrate targets.
The authors make claims about the geotropisms of these two tick species without citations. It cannot be stated that I. scapularis is “negatively geotropic” without a peer reviewed source. Additionally, I. scapularis is known to exhibit very different questing behavior depending on whether southern or northern populations are being studied. More caution should be exercised in this paragraph.
Sit and wait – consider replacing with “ambush”
“I. scapularis did not show interest in the thermal cue on the flying “T”, possibly because it was still looking for an appropriate place to sit and wait” . Absolutely not! I. scapularis will immediately settle and feed in the exact spot it is placed on an ideal host such as a shaved tick-naïve guinea pig, and certain locations like the ears of a mouse or rabbit. There are more subtle stimuli at work here. Please use caution with blanket statements about tick behavior.
Author Response
- Reviewer 4: The manuscript written by Otalora luna et al. is interesting and nicely written.
R: Thank you.
- I am unable to provide a sufficient review for some of the methods and calculations relating to tick orientation. Additionally, no line numbers were included in the draft provided, therefore, I have not matched line numbers with suggested edits. I strongly recommend that authors revise commentary on I. scapularis behavior in the Discussion.
R: We understand; we rephrase such commentary.
- Intro - Need to cite a reference for expansion of I. scapularis in North America… Eisen or Ogden, or numerous others. This is occurring both northward and westward.
R: We cited more references for expansion of both species, including Eisen and Ogden works. We underlined that his is occurring at diverse locations in North America.
- Amblyomma americanum and tularemia….yes, a vector in a few extremely rare instances. Practically, I would suggest replacing this pathogen/disease with Ehrlichia ewingii, Rickettsia or Heartland Virus.
R: We included these and excluded tularemia.
- americanum is missing an ‘I’ (spelling in the paragraph preceding Methods section
R: Thank you.
- Methods 2.1 – see formatting gap
R: We improved the formatting.
- Figure 2A is partially obscured, making it impossible to review
R: We upload a high quality figure, probably this happened during the post-submission process.
- Figure 3 – does this represent the track of a single tick? If not, the number should be listed in the Legend. Is this figure mentioned in the text? If not, it’s purpose for inclusion should be stated.
R: Figure 3 represents the track of two single ticks (we edited the legend), and it was mentioned in the text, in results.
- One key element the authors may want to address is the lack of certainty around what the range of intensities of heat source that are recognizable to the tick and how much that influences potential attraction. However, heat sources in nature relative to thermal conditions are also variable, so this is just a consideration and limitation of these methods, but not a fault.
R: We addressed this at the end of the discussion; thank you.
- The results that linearity was not affected is surprising and suggests that ticks, at least the more ambulatory Amblyomma were not positioned for enough from the stimuli?
R: As can be seen in Fig 3, A. amercianum linearity was clearly affected by stimulation, but we couldn’t demonstrate statistical significance for other individuals.
- Dermacentor variabilis misspelling in discussion
R: Thank you.
- These two tick species aren’t really known to “drop” on hosts as stated in the Discussion. In nature, they clasp on to hosts passing in close proximity. They do not ascend vegetation higher than the height of vertebrate targets.
R: We deleted “drop”.
- The authors make claims about the geotropisms of these two tick species without citations. It cannot be stated that I. scapularis is “negatively geotropic” without a peer reviewed source. Additionally, I. scapularis is known to exhibit very different questing behavior depending on whether southern or northern populations are being studied. More caution should be exercised in this paragraph.
R: This is Otálora-Luna et al. unpublished, which has been accepted by Journal of Ethology, and might be citable soon. If not, we will exclude this comment.
- Sit and wait – consider replacing with “ambush”
R: We included ambush.
- “I. scapularis did not show interest in the thermal cue on the flying “T”, possibly because it was still looking for an appropriate place to sit and wait” . Absolutely not! scapularis will immediately settle and feed in the exact spot it is placed on an ideal host such as a shaved tick-naïve guinea pig, and certain locations like the ears of a mouse or rabbit. There are more subtle stimuli at work here. Please use caution with blanket statements about tick behavior.
R: We agreed. We modified this sentence.
Round 2
Reviewer 2 Report
The Authors improved their manuscript. In my opinion ms can be considered for publication in Insects
Author Response
Thank your very much!
Reviewer 3 Report
Improvements sufficient. Nice work.
Author Response
Thank you very much!